

# Kernel polynomial method for linear spin wave theory

Harry Lane[1,*,°], Hao Zhang[2], David Dahlbom[3], Sam Quinn[4], Rolando D. Somma[5],
Martin Mourigal[4], Cristian D. Batista[3,6] and Kipton Barros[2,†]

**1** School of Physics and Astronomy, University of St Andrews, St Andrews, United Kingdom
**2** Theoretical Division and CNLS, Los Alamos National Laboratory, Los Alamos, United States
**3** Neutron Scattering Division, Oak Ridge National Laboratory, Oak Ridge, United States
**4** School of Physics, Georgia Institute of Technology, Atlanta, United States
**5** Google Quantum AI, Venice, United States
**6** Department of Physics and Astronomy, The University of Tennessee,
Knoxville, United States

★ htl7@st-andrews.ac.uk , † kbarros@lanl.gov

## Abstract

Calculating dynamical spin correlations is essential for matching model magnetic exchange Hamiltonians to momentum-resolved spectroscopic measurements. A major numerical bottleneck is the diagonalization of the dynamical matrix, especially in systems with large magnetic unit cells, such as those with incommensurate magnetic structures or quenched disorder. In this paper, we demonstrate an efficient scheme based on the kernel polynomial method for calculating dynamical correlations of relevance to inelastic neutron scattering experiments. This method reduces the scaling of numerical cost from cubic to linear in the magnetic unit cell size.



## Contents

---

°Present address: University of Manchester at Harwell, Harwell Campus, Didcot, Oxfordshire OX11 0FA, United Kingdom.

# 1 Introduction

The existence of coherent low-energy spin wave excitations in ordered spin systems has been known since the first half of the 20$^\text{th}$ century. The thermal occupation of these low energy modes gives rise to the $T^{3/2}$ scaling of the magnetization as suggested by Bloch [1]. A highly successful and more complete theoretical treatment of spin waves was provided in later work by Dyson [2], Anderson [3], Kubo [4] and Holstein and Primakoff [5]. In many cases, considering only harmonic fluctuations and neglecting interaction terms is sufficient and corresponds to linearizing the equation of motion governing the dynamics of the spin operators' expectation values. This treatment is exact in the limit $1/S \to 0$.

The utility of linear spin wave theory (LSWT) to describe experimental results is fully displayed in neutron spectroscopy. The often quantitative match between spin wave theory and inelastic neutron scattering from magnetic insulators enables the inverse modeling of measured neutron spectra to extract the parameters of exchange Hamiltonians. This inverse modeling is not merely restricted to fitting the dispersion relations. The entire dynamical correlation functions and intrinsic line widths offer deep insight into spatial anisotropies and finite lifetime effects, respectively. This insight has been instrumental across many topics in quantum materials, from understanding the spin correlations in cuprate and Fe-based superconductors [6–8] to searching for quantum spin liquid candidates [9–12].

Inelastic neutron scattering experiments on modern spectrometers can collect data on a dense grid of billions of pixels in the four-dimensional space $(\mathbf{q}, \hbar\omega)$. For inverse modeling, it is typically necessary to integrate over several dimensions and perform fits of binned data in two or three dimensions. Accounting for this binning and instrument resolution is expensive but sometimes vital [13]. With advances in machine learning [14], there is hope that the highly sensitive determination of model Hamiltonians from large volumes of spectroscopic data can be accelerated [15–19]. This sensitivity is likely key to determining the spin Hamiltonians of important quantum material systems such as $\alpha$-RuCl$_3$ [16,20] where a flat parameter space has led to wildly different predictions of Hamiltonian parameters.

The development of efficient forward calculators is key to leveraging powerful novel optimization methods with large quantities of input data. Several efficient implementations of LSWT exist such as McPhase [21,22], SpinW [23], PyLiSW [24], SpinWave [25] and SpinWaveGenie [26]. These leverage the traditional spin wave theory workflow, which amounts to finding the non-interacting magnon spectrum through diagonalization of the dynamical matrix [27–29]. The cost of this operation scales as $\sim \mathcal{O}(N^3)$ in the system size $N$. This makes LSWT modeling prohibitively expensive for systems with large magnetic unit cells, including incommensurate, multi-$\boldsymbol{k}$ and disordered systems. In this paper, we outline a new approach based on the kernel polynomial method [30], which reduces the computational complexity of calculating the dynamical correlations to $\sim \mathcal{O}(N)$ in system size. We then demonstrate significant practical speedups, relative to convention LSWT implementations, for several physically relevant systems. We anticipate that this improved computational efficiency will be vital to the inverse modeling of spectroscopic measurements for systems with especially large magnetic unit cells.

## 2 Review of linear spin wave theory

There are a number of formal steps to arrive at the LSWT predictions for dynamical correlations. The starting point is a quantum spin Hamiltonian. The naïve replacement of operators with expectation values yields a classical Hamiltonian. Minimization of the classical Hamiltonian with respect to expectation values yields a magnetically ordered ground state. LSWT considers expansion of the quantum operators about the assumed magnetic ordering, under the assumption of small fluctuations. Each local observable operator may be represented as a quadratic form of Schwinger bosons [31]. In the conventional approach, only the spin-dipole is retained, yielding two bosonic flavors. For sites with spin-$S$, a more variationally accurate approximation employs $N = 2S + 1$ bosonic flavors per quantum spin [32]. An associated classical limit of SU($N$) coherent spin states has recently been derived [33], along with efficient numerical techniques [34,35]. The contributions in this paper will apply independently of the selected variant of the LSWT (dipole-only, or multi-flavor bosons). It should be noted that the traditional, dipole-only LSWT can be made more accurate through a renormalization procedure that implicitly accounts for the spin multipoles [36]. This allows, for example, to directly calculate the expected energy of a given single-ion anisotropy $\langle \vec{\Omega} | (\hat{S}_i^z)^2 | \vec{\Omega} \rangle$, while avoiding the approximation $\langle \vec{\Omega} | \hat{S}_i^z | \vec{\Omega} \rangle \langle \vec{\Omega} | \hat{S}_i^z | \vec{\Omega} \rangle$, which would only be valid when $S \to \infty$. Here, $| \vec{\Omega} \rangle$ denotes an SU($N$) coherent spin state that is a pure dipole, $\vec{\Omega}$.

It is convenient to work in a local reference frame such that the $z$-axis for each site is aligned with the magnetic ordering. Because fluctuations must be perpendicular to the local $z$-axis, one can perform a "condensation" along this axis, yielding the Holstein-Primakoff (HP) bosons $\hat{\mathbf{a}}^\dagger = (\hat{\boldsymbol{\alpha}}^\dagger, \hat{\boldsymbol{\alpha}})$, here written in Nambu spinor-like form. Like the Schwinger bosons, these HP bosons satisfy canonical commutation relations. If the system has translation invariance, then the quasi-momentum $\mathbf{q}$ becomes a good quantum number, and the Hamiltonian can be diagonalized in blocks. Conventionally, one would work with $\hat{\mathbf{a}}_{\mathbf{q}}^\dagger = (\hat{\boldsymbol{\alpha}}_{\mathbf{q}}^\dagger, \hat{\boldsymbol{\alpha}}_{-\mathbf{q}})$. In our presentation, for pedagogical simplicity, we omit the $\mathbf{q}$ index and assume that $\hat{\boldsymbol{\alpha}}$ becomes an abstract list of $n$ operators, with $\hat{\boldsymbol{\alpha}}^\dagger$ their Hermitian conjugates. This simplification can be readily justified in real space, and also in Fourier space if we assume enlarged spinor objects, $\hat{\mathbf{a}}_{\mathbf{q}}^\dagger = ((\hat{\boldsymbol{\alpha}}_{\mathbf{q}}^\dagger, \hat{\boldsymbol{\alpha}}_{-\mathbf{q}}^\dagger), (\hat{\boldsymbol{\alpha}}_{\mathbf{q}}, \hat{\boldsymbol{\alpha}}_{-\mathbf{q}}))$. Note that each $\hat{\boldsymbol{\alpha}}_{\mathbf{q}}$ includes an index over sites of the magnetic unit cell. When generalizing to a theory of multi-flavor bosons, the $\hat{\boldsymbol{\alpha}}_{\mathbf{q}}$ will also carry a flavor index [32].

The original spin Hamiltonian is next expanded in powers of the HP bosons, which are associated with "small" fluctuations away from the classical ground state. At zeroth order, one recovers the classical energy $H$. The linear order term should vanish, provided that the magnetically ordered state is indeed a local minimum of the classical energy $H$. The most general Hamiltonian at quadratic order takes the form,

$$\hat{\mathcal{H}} = \frac{1}{2}\hat{\mathbf{a}}^{\dagger} D \hat{\mathbf{a}} = \frac{1}{2}(\hat{\boldsymbol{\alpha}}^{\dagger}, \hat{\boldsymbol{\alpha}}) \begin{pmatrix} A & B \\ B^* & A^* \end{pmatrix} \begin{pmatrix} \hat{\boldsymbol{\alpha}} \\ \hat{\boldsymbol{\alpha}}^{\dagger} \end{pmatrix}, \tag{1}$$

and this will be the starting point for the LSWT calculation. Higher order terms in the expansion of the spin Hamiltonian would be needed for perturbative corrections to LSWT, but that is outside the scope of the present work. Interested readers are referred to Ref. [37] for an in-depth discussion of nonlinear corrections to LSWT.

Because $\hat{\mathcal{H}}$ is Hermitian, the $2n \times 2n$ matrix $D$ must be also. This implies that the $n \times n$ sub-matrix $A$ is Hermitian, while the sub-matrix $B$ is symmetric, but not necessarily Hermitian. The notation $A^*$ and $B^*$ denotes element-wise complex conjugation, *without* matrix transpose. The matrix $D$ will be positive semi-definite, because it originates in an expansion about an energy minimum. In practice, to avoid divergences, we will further require that $D$ is positive *definite*. To study zero-energy modes, one might impose a shift $D \to D + \epsilon$ and then take the limit $\epsilon \to 0^+$.

In the traditional LSWT, the next step is to bring the Hamiltonian into diagonal form, $\hat{\mathcal{H}} = \hat{\mathbf{b}}^{\dagger} \Omega \hat{\mathbf{b}}/2$. For this, one seeks a transformation from the HP bosons to the Bogoliubov bosons,

$$\hat{\mathbf{b}} = T^{-1}\hat{\mathbf{a}}, \tag{2}$$

that satisfies

$$T^{\dagger} D T = \Omega, \tag{3}$$

for some diagonal matrix $\Omega$. The components of $\hat{\mathbf{b}}$ must be bosonic operators that satisfy canonical commutation relations. The conditions for $T$ to be canonical are reviewed in Appendix A. In particular, $T$ must be para-unitary,

$$T^{-1} = \tilde{I} T^{\dagger} \tilde{I}, \tag{4}$$

or equivalently $T\tilde{I}T^{\dagger} = T^{\dagger}\tilde{I}T = \tilde{I}$, where

$$\tilde{I} = \text{diag}(1, \ldots, 1, -1, \ldots, -1), \tag{5}$$

is a $2n \times 2n$ diagonal matrix,

Assuming para-unitarity, Eq. (3) becomes a diagonalization of the non-Hermitian matrix,

$$\tilde{I} D = T(\tilde{I}\Omega)T^{-1}. \tag{6}$$

This coincides with the generalized eigenvalue problem,

$$\tilde{I}\mathbf{t}_j = \lambda_j D \mathbf{t}_j, \tag{7}$$

where $\mathbf{t}_j$ are the columns of $T$, and $\lambda_j$ are the diagonal elements of $(\tilde{I}\Omega)^{-1}$. Recall that $\tilde{I}$ and $D$ are Hermitian matrices, with $D$ positive definite by assumption. A theorem of linear algebra states that a complete set of generalized eigenvectors $\mathbf{t}_j$ exists, corresponding to real eigenvalues $\lambda_j$. The numerical linear algebra package LAPACK [38] provides a subroutine ZHEGV to directly calculate all generalized eigenpairs $(\mathbf{t}_j, \lambda_j)$. A common implementation uses the Cholesky decomposition of $D$ [27].

Consider the anti-unitary operator $J$ defined by the action

$$J\begin{pmatrix} \mathbf{r} \\ \mathbf{s}^* \end{pmatrix} = \begin{pmatrix} \mathbf{s} \\ \mathbf{r}^* \end{pmatrix}. \tag{8}$$

Given the $2 \times 2$ block structure of $D$, if $(\mathbf{t}_j, \lambda_j)$ is an eigenpair for Eq. (7), then $(J\mathbf{t}_j, -\lambda_j)$ is also an eigenpair. To make $T$ a canonical transformation, it is necessary to order the eigenvalues such that,

$$\tilde{I}\Omega = \mathrm{diag}(E_1, \dots E_n, -E_1, \cdots -E_n), \tag{9}$$

where $E_j = \lambda_j^{-1}$ is positive for $j = 1, \dots, n$. This ordering ensures that $\Omega$ is strictly positive, and

$$T = [\mathbf{t}_1 \dots \mathbf{t}_n, J\mathbf{t}_1 \dots J\mathbf{t}_n]. \tag{10}$$

Referring to Eq. (7), one finds $\mathbf{t}_j^\dagger \tilde{I} \mathbf{t}_k \propto \tilde{I}_{jk}$ with positive constant of proportionality because $\mathbf{t}_j^\dagger D \mathbf{t}_j > 0$ by assumption on $D$. Therefore, with suitable normalization of $\mathbf{t}_j$, the para-unitary condition, $T^\dagger \tilde{I} T = \tilde{I}$, is achieved. Referring to Appendix A, these results establish that the Bogoliubov bosons $\hat{\mathbf{b}}^\dagger = (\hat{\boldsymbol{\beta}}^\dagger, \hat{\boldsymbol{\beta}})$ satisfy canonical commutation relations.

In summary, diagonalizing the matrix $\tilde{I}D$ produces a canonical transformation $T$ that maps between HP and Bogoliubov bosons, and the latter bring the quantum Hamiltonian to diagonal form. Using the positivity of $\Omega$ and the bosonic commutation relation, $\beta_j \beta_j^\dagger = \beta_j^\dagger \beta_j + 1$, the result is,

$$\hat{\mathcal{H}} = \sum_{j=1}^{n} E_j(\hat{\beta}_j^\dagger \hat{\beta}_j + 1/2), \tag{11}$$

where each Bogoliubov boson operator $\hat{\beta}_j^\dagger$ creates a quasi-particle excitation of energy $E_j$.

Dynamical correlations can now be formulated. Let $\hat{A}$ and $\hat{B}$ denote arbitrary physical observables. Consistent with our assumption that fluctuations are small, each operator may be approximated as a linear combination of the HP bosons,

$$\hat{A} = \hat{\mathbf{a}}^\dagger \mathbf{u}, \qquad \hat{B} = \hat{\mathbf{a}}^\dagger \mathbf{v}, \tag{12}$$

for appropriate complex vectors $\mathbf{u}$ and $\mathbf{v}$ with $2n$ components. If the operators $\hat{A}$ and $\hat{B}$ represent local observables defined in real space, then $\hat{A}^\dagger = \hat{A}$ and $\hat{B}^\dagger = \hat{B}$. Alternatively, if they have well-defined momentum $\mathbf{q}$, they transform as $\hat{A}_\mathbf{q}^\dagger = \hat{A}_{-\mathbf{q}}$. The latter case is relevant for computing inelastic scattering cross sections.

Now consider the dynamical correlation in thermal equilibrium,

$$C_\omega^{B^\dagger, A} = \sum_{\nu, \mu} \langle \nu | \hat{B}^\dagger | \mu \rangle \langle \mu | \hat{A} | \nu \rangle \frac{e^{-\beta \epsilon_\nu}}{\mathcal{Z}} \delta(\epsilon_\mu - \epsilon_\nu - \omega), \tag{13}$$

where $\mathcal{Z} = \sum_\mu e^{-\beta \epsilon_\mu}$ is the (grand canonical) partition function, and the indices $\mu$ and $\nu$ label eigenstates $|\mu\rangle$ and $|\nu\rangle$ of the quadratic Hamiltonian, $\hat{\mathcal{H}}|\mu\rangle = \epsilon_\mu |\mu\rangle$. Each eigenstate $|\mu\rangle = |N_1 N_2 \dots N_n\rangle$ is labeled by its Bogoliubov boson occupation numbers $N_j$, and has energy $\epsilon_\mu = \sum_{j=1}^{n} N_j E_j$.

The principle of detailed balance is a general consequence of (13) and states,

$$C_{-\omega}^{B^\dagger, A} = e^{-\beta \omega} C_\omega^{A, B^\dagger}. \tag{14}$$

That is, the negative frequency result can be obtained from a *different* correlation function at positive frequency.

For the remainder of this paper, it suffices to consider $\omega > 0$. The correlation function $C_\omega \equiv C_\omega^{B^\dagger, A}$ can be calculated explicitly (see Appendix B),

$$C_\omega = [1 + n_{\mathrm{B}}(\omega)] \sum_{j=1}^{n} \mathbf{v}^\dagger \mathbf{t}_j \mathbf{t}_j^\dagger \mathbf{u}\, \delta(\omega - E_j) \qquad (\omega > 0), \tag{15}$$

which involves the Bose function,

$$n_{\mathrm{B}}(\omega) = (e^{\beta \omega} - 1)^{-1}. \tag{16}$$

This completes a traditional recipe for calculating intensities within the framework of LSWT. Given a quadratic bosonic Hamiltonian, Eq. (1), the steps are:

1. Solve the generalized eigenvalue problem of Eq. (7) to calculate a para-unitary transform $T$ that maps from $\mathbf{a}^\dagger$ to the Bogoliubov bosons $\mathbf{b}^\dagger$. The latter represent non-interacting quasi-particle excitations with energies $E_1, \ldots, E_n$, as in Eq. (11).

2. Calculate intensities associated with the dynamical correlation for bands $j = 1 \ldots n$ using Eq. (15). The operators $\hat{A}$ and $\hat{B}$ may be arbitrary, but are approximated as a linear combination of HP bosons, involving the $2n$-component complex vectors $\mathbf{u}$ and $\mathbf{v}$, Eq. (12).

This procedure is widely deployed for studying systems with translation invariance and small magnetic unit cells [23]. In some application contexts, however, the explicit calculation of $T$ may become a numerical bottleneck. For example, very large effective magnetic unit cells will arise in studies of systems with quenched disorder, systems with long-wavelength magnetic modulations, or in (approximations to) systems with multiple incommensurate wave-vectors. The computational cost to explicitly diagonalize $\tilde{I}D$ will scale cubically in the volume of the magnetic unit cell. Further, to calculate momentum-dependent correlation functions, such as the components of the spin structure factor $\mathcal{S}^{\alpha\beta}(\mathbf{q}, \omega) \equiv C_\omega^{S_\mathbf{q}^{\alpha\dagger}, S_\mathbf{q}^\beta}$, one must employ an *independent* LSWT calculation procedure for each wavevector $\mathbf{q}$ of interest, and this can lead to a very large overall computational cost. In the following sections, we detail an algorithm that significantly accelerates the calculation for large magnetic unit cells.

## 3 Acceleration of LSWT using the kernel polynomial method

The kernel polynomial method (KPM) allows estimation of matrix properties while avoiding explicit matrix diagonalization [30]. This approach has been used in a diverse range of physics contexts including the Kondo lattice model [39,40], non-Hermitian Hamiltonians [41], many-body localization [42, 43] and flat-band physics in semimetals [44, 45]. In this work, we explore the application of KPM to LSWT; this reduces scaling of computational cost from cubic to linear in the volume of the magnetic unit cell. Such acceleration can be achieved provided that the dynamical matrix $D$ is sparse, or otherwise can be efficiently applied to a given vector. An alternative technique to measure dynamical correlations is integration of the classical equations of motion, commonly referred to as Landau-Lifshitz Dynamics (LLD), which also scales linearly in system size [46]. Advantages of KPM-LSWT (henceforth simply referred to as KPM) over classical dynamics at small $k_B T$ are as follows:

1. Through periodic extension of the magnetic unit cell, there is no restriction that the $\mathbf{q}$ vectors be commensurate with a given system size. For example, one can calculate $\mathcal{S}(\mathbf{q}, \omega)$ for an arbitrary path in reciprocal space. The cost scales linearly in both the size of the magnetic unit cell, and also in the number of selected $\mathbf{q}$ vectors.

2. There is no stochastic error associated with averaging over randomly sampled equilibrium conditions of the magnetic configuration.

3. The prefactor to the linear-scaling computational cost can be directly controlled by reducing resolution in the energy, $\omega$.

It should be emphasized that the method outlined in this paper considers only small fluctuations about a magnetic ground state and is valid at low temperatures. If the temperature is comparable with the Hamiltonian parameters, $k_B T \sim J$, then nonlinearities associated with thermal fluctuations become relevant, and an approach such as LLD with coupling to a Langevin thermostat should be used instead [46].

## 3.1 Intensities without matrix diagonalization

Our aim is to express Eq. (15) in a compact form that avoids explicit reference to the matrix diagonalization of Eq. (6).

The dynamical correlation function can be written

$$C_\omega = [1 + n_{\mathrm{B}}(\omega)]\mathbf{v}^\dagger \rho_+(\omega)\mathbf{u} \qquad (\omega > 0). \tag{17}$$

The matrix-valued distribution,

$$\rho_+(\omega) = \sum_{j=1}^{n} \mathbf{t}_j\, \delta(\omega - E_j)\mathbf{t}_j^\dagger, \tag{18}$$

will play a key role in what follows. It can be viewed as a Green's function associated with only the *positive* eigenvalues $E_j$ of the matrix $\tilde{I}D$. Referring to Eqs. (6) and (9), the eigenvalues of $\tilde{I}D$ come in pairs, $\pm E_j$. We can extend the summation to all eigenvalues by introducing,

$$g_\omega(x) = \Theta(\omega)\delta(\omega - x), \tag{19}$$

with $\Theta(\cdot)$ being the Heaviside step function. This allows us to write,

$$\rho_+(\omega) = \sum_{j=1}^{2n} \mathbf{t}_j\, g_\omega(\tilde{I}_{jj}\Omega_{jj})\tilde{I}_{jj}\mathbf{t}_j^\dagger, \tag{20}$$

where only the terms $j = 1,\ldots,n$ actually contribute the sum and the negative eigenvalues are zeroed out by the Heaviside step function.

Because $\tilde{I}$ and $\Omega$ are each diagonal, it will be possible to simplify this expression by introducing matrix notation. The key idea is to view $g_\omega(\cdot)$ as a matrix function that maps from the diagonal matrix $\tilde{I}\Omega$ to a new diagonal matrix $g_\omega(\tilde{I}\Omega)$.[1] In particular,

$$g_\omega(\tilde{I}\Omega)_{jk} = g_\omega(\tilde{I}_{jj}\Omega_{jj})\delta_{jk}. \tag{21}$$

Then the sum over the $j$ index can be implemented as matrix multiplication. Explicitly,

$$\rho_+(\omega) = \sum_{j,k,\ell=1}^{2n} \mathbf{t}_j\, g_\omega(\tilde{I}\Omega)_{jk}\tilde{I}_{k\ell}\mathbf{t}_\ell^\dagger = T g_\omega(\tilde{I}\Omega)\tilde{I}T^\dagger. \tag{22}$$

Using the para-unitary condition of Eq. (4) and the identity $\tilde{I}^2 = I$,

$$\rho_+(\omega) = T g_\omega(\tilde{I}\Omega)T^{-1}\tilde{I}. \tag{23}$$

---

[1]Although $g_\omega(\cdot)$ is formally defined as matrix-valued distribution, one can imagine constructing it from *smoothed* Dirac-$\delta$ functions, to any desired level of approximation.

Recall that matrix functions are defined by their action in the diagonal basis, via the transformation of eigenvalues. From the diagonalization of Eq. (6), it follows,

$$g_\omega(\tilde{I}D) = T g_\omega(\tilde{I}\Omega) T^{-1}. \tag{24}$$

Substituting yields our final result,

$$\rho_+(\omega) = \Theta(\omega)\delta(\omega - \tilde{I}D)\tilde{I}. \tag{25}$$

Crucially, the right-hand side no longer makes explicit reference to the eigenvectors or eigenvalues of the matrix $\tilde{I}D$. With this reformulation, one can avoid direct matrix diagonalization, and instead use polynomial approximation techniques via expansion in powers of $\tilde{I}D$.

If we can efficiently estimate the matrix-vector product $\rho_+(\omega)\mathbf{u}$, then the desired dynamical correlations $C_\omega$ follow directly from Eq. (17). One possibility is to apply matrix-polynomial approximation to obtain a smoothed estimate of $\rho_+(\omega)$. This direct approach is viable, and is presented in Appendix D. We argue, however, that it is better to first incorporate known broadening effects that will effectively smooth the intensities $C_\omega$, leading to faster convergence in the polynomial expansion.

## 3.2 Broadening and regularization

Experimental measurements of $C_\omega$ will have broadened peaks, rather than the idealized Dirac-$\delta$ peaks in Eq. (15). An effective broadening kernel $G_0(\cdot, \cdot)$ will arise from limitations in the experimental energy resolution and from intrinsic quantum effects that are beyond LSWT. These effects are commonly modeled as Gaussian and Lorentzian lineshapes, respectively. The two lineshapes can be combined via convolution, yielding a Voigt function. To reduce cost, this is frequently approximated as a pseudo-Voigt function. Note that the broadening line-width $\sigma$ may also be energy dependent. All of these effects can be captured by some appropriate choice of kernel $G_0(\cdot, \cdot)$.

Our numerical aim is to calculate smoothed intensities,

$$\tilde{C}_\omega = \int_{-\infty}^{\infty} G(\omega, \omega') C_{\omega'} d\omega'. \tag{26}$$

To make contact with Eq. (17), we focus on the case where intensities originate from positive quasi-particle energies, $\omega' > 0$. If desired, one may still account for sources $\omega' < 0$ by performing a separate calculation, as indicated by detailed balance, Eq. (14). To restrict the integration domain to $\omega' > 0$, we are employing,

$$G(\omega, \omega') = \vartheta(\omega') G_0(\omega, \omega'), \tag{27}$$

where $G_0(\omega, \omega')$ is the desired broadening kernel, and $\vartheta(\omega')$ is an empirical function that masks contributions from negative $\omega'$. Then $G(\omega, \omega')$ can be viewed as a smoothed version of $g_\omega(\omega')$, Eq. (19).

To simplify the expression of $\tilde{C}_\omega$, introduce a new function associated with the broadened intensity at energy transfer $\omega$, as originated from a positive quasi-particle energy $\omega' = x$,

$$f_\omega(x) = [1 + n_B(x)] G(\omega, x). \tag{28}$$

Because of the implicit $\vartheta(x)$ factor, $f_\omega(x) = \Theta(x) f_\omega(x)$ by construction. Using this identity, and also substituting from Eqs. (17) and (25), one may calculate,

$$\tilde{C}_\omega = \int_{-\infty}^{\infty} f_\omega(x) \mathbf{v}^\dagger \rho_+(x) \mathbf{u} \, dx$$
$$= \mathbf{v}^\dagger \left[ \int_{-\infty}^{\infty} f_\omega(x) \delta(x - \tilde{I}D) \, dx \right] \tilde{I} \mathbf{u}, \tag{29}$$

which integrates to,[2]

$$\tilde{C}_\omega = \mathbf{v}^\dagger f_\omega(\tilde{I}D)\tilde{I}\mathbf{u}. \tag{30}$$

Our results so far hold for any $\vartheta(x)$ that masks contributions from negative $x$. For purposes of polynomial approximation, there are benefits to selecting $\vartheta(x)$ smooth. A good option is

$$\vartheta(x) = \begin{cases} 0, & x < 0, \\ (4 - \frac{3x}{\sigma'})\frac{x^3}{\sigma'^3}, & 0 \le x \le \sigma', \\ 1, & x > \sigma', \end{cases} \tag{31}$$

which broadens the step function over a tunable energy scale $\sigma'$. In effect, $\vartheta(x)$ dampens intensities that may arise from low-energy modes, $x \lesssim \sigma'$. The scaling $\vartheta(x) \sim 4x^3/\sigma'^3$ at small $x$ is designed to counteract the singularity of the Bose function $n_{\mathrm{B}}(x) \sim 1/\beta x$. Consequently, $f_\omega(x)$ and its first derivative will be continuous everywhere. This smoothness makes $f_\omega(x)$ well suited for polynomial approximation, to be discussed in the next section.

The energy cut-off scale $\sigma'$ is an empirical parameter. If the unbroadened spectrum $C_\omega$ is known to have a quasi-particle gap, then its size is a natural choice for $\sigma'$. Otherwise, one must be aware that some spectral intensity can be lost in the approximation $\vartheta(x) \approx \Theta(x)$. Making this sacrifice, however, can bring numerical benefits. As we will describe in the next section, the overall numerical cost for a linear-scaling spin wave calculation will be inversely proportional to the smallest energy resolution scale, $\min(\sigma, \sigma')$, where $\sigma$ is the smallest characteristic linewidth of the broadening kernel $G_0(\cdot, \cdot)$.

## 3.3 Fast polynomial approximation

Numerical evaluation of the intensities in Eq. (30) will require calculating the matrix-vector product $f_\omega(\tilde{I}D)\tilde{I}\mathbf{u}$. KPM enables this calculation without explicit construction of the dense matrix $f_\omega(\tilde{I}D)$. Instead, the method requires only that the dynamical matrix $D$ can be efficiently applied to a given vector. A brief review of KPM is presented in Appendix C. Here, we state the final procedure.

To apply KPM, first construct a scaled matrix

$$A = \tilde{I}D/\gamma, \tag{32}$$

where $\gamma$ is defined such that all eigenvalues of $A$ lie between $-1$ and $1$. A generalization of the Lanczos algorithm [47,48] to $\eta$-pseudo-Hermitian matrices [49], can be used to estimate bounds on the extremal eigenvalues of $\tilde{I}D$, which determines a valid scaling $\gamma$.

The key idea in KPM is to approximate the unknown function as a finite-order matrix polynomial,

$$f_\omega(\gamma A) \approx \sum_{m=0}^{M-1} c_{m,\omega} T_m(A). \tag{33}$$

The order-$m$ Chebyshev polynomial can be written $T_m(x) = \cos(m \arccos x)$ when $-1 \le x \le 1$. This identity establishes a close relationship between the Chebyshev and Fourier series. The coefficients,

$$c_{m,\omega} = \frac{1}{q_m} \int_{-1}^{+1} w(x) T_m(x) f_\omega(\gamma x) dx, \tag{34}$$

---

[2]To avoid the matrix-valued Dirac $\delta$-function, one could instead substitute $\rho_+$ from Eq. (20). Then integration yields a summation over eigenvalues of $f_\omega(\tilde{I}D)$, from which the same matrix expression can be reconstructed.

are analogous to a Fourier transform of $f_\omega(\cdot)$. Here, $w(x) = (1 - x^2)^{-1/2}$, $q_0 = \pi$, and $q_{m \geq 1} = \pi/2$. Given $\omega$, the coefficients $c_{m,\omega}$ for all $m = 0$ to $M - 1$ are efficiently evaluated using Chebyshev-Gauss quadrature and the discrete cosine transform. The procedure is described in Appendix C.2.

The higher the polynomial degree $M$, the better the ability to approximate sharp features in $f_\omega(\cdot)$. If $\sigma$ denotes the target energy resolution scale (e.g., the characteristic width of the line broadening kernel), then an appropriate polynomial order is $M = c\gamma/\sigma$, where the constant $c$ is typically of order 5–10. Beyond this, the error decays approximately exponentially in $c$. In the examples of section 4, we find that $M \lesssim 350$ is typically sufficient for an LSWT calculation.

To achieve linear scaling in computational cost, it is necessary to avoid explicit construction of the dense matrices $T_m(A)$. Instead, we work with the vectors,

$$\varphi_m = T_m(A)\tilde{I}\mathbf{u}. \tag{35}$$

These are efficiently calculated using a two-term recurrence associated with the Chebyshev polynomials,

$$\varphi_0 = \tilde{I}\mathbf{u}, \tag{36}$$

$$\varphi_1 = A\tilde{I}\mathbf{u}, \tag{37}$$

$$\varphi_{m+1} = 2A\varphi_m - \varphi_{m-1}. \tag{38}$$

After each vector $\varphi_m$ is calculated, one can obtain the scalar dot product,

$$\mu_m = \mathbf{v}^\dagger \varphi_m = \mathbf{v}^\dagger T_m(A)\tilde{I}\mathbf{u}. \tag{39}$$

Given the moments $\mu_m$ for $m = 0$ to $M - 1$, it becomes possible to estimate the dynamical correlations of Eq. (30),

$$\tilde{C}_\omega = \mathbf{v}^\dagger f_\omega(\gamma A)\tilde{I}\mathbf{u} \approx \sum_{m=0}^{M-1} c_{m,\omega}\mu_m. \tag{40}$$

For large matrices $A$, calculating all the moments $\mu_m$ will be the dominant numerical expense. These moments should be calculated only once, independently of $\omega$. Evaluating the sum of Eq. (40) is an additional cost that is independent of system size. To perform this sum quickly, one should precalculate the dense matrix of coefficients $c_{m,\omega}$. The Chebyshev moments, for each $\mathbf{q}$ of interest, may be collected into another dense matrix, $\mu_{m,\mathbf{q}}$. Then the contraction over $m$ may be recognized as dense matrix-matrix multiplication, which efficiently yields the intensities $\tilde{C}_{\omega,\mathbf{q}}$.

Sometimes it will be desired to calculate the dynamical correlations between an operator $\hat{A}$ and multiple operators $\hat{B}_1, \hat{B}_2, \ldots$. Each observable $\hat{B}_i$, labeled by the index $i$, is associated with a different vector $\mathbf{v}_i$. The dynamical correlation functions are,

$$\tilde{C}_{i,\omega} \approx \sum_{m=0}^{M-1} c_{m,\omega}\mu_{i,m}, \tag{41}$$

where now the moments $\mu_{i,m} = \mathbf{v}_i^\dagger \varphi_m$ carry an $i$ index. The $\tilde{C}_{i,\omega}$ can be calculated for all $i$ using only a single Chebyshev recursion, Eq. (38). The key observation is that the vectors $\varphi_m$ are independent of $i$. Upon calculating each $\varphi_m$, it is fast to perform many vector dot products, yielding the moments $\mu_{i,m}$ for all $i$. In the context of linear spin wave theory, the index $i$ will run over all combinations of $\alpha, \beta \in \{x, y, z\}$ forming all components of the dynamical structure factor.

# 4 Example KPM calculations

We now present several examples on systems for which the large unit cell size makes conventional LSWT impractical. We focus on three examples, each possessing a property that leads to a large effective unit cell, and compare the time required to compute the spectrum with that of the comparable LSWT calculation. In each example, the mean-field ground state is first found by simulated annealing followed by gradient descent. The spectra are plotted on a grid in $(\mathbf{q}, \hbar\omega)$ through several high symmetry points [50, 51]. The spectrum are plotted symmetrically such that the KPM result and the LSWT result are mirror-symmetric about the center of the figure (chosen to be the $\Gamma$-point). As noted in section 3, $M = c\gamma/\sigma$ with $c \sim$ 5-10 is typically sufficient for convergence. To determine a cutoff order in the Chebyshev expansion more systematically, we take 50 energies spanning $[2\sigma, \hbar\omega_{max}]$ at 50 random $\mathbf{q}$ positions in the first Brillouin zone and calculate the spectrum with different truncation orders, $M$ in steps of 25. We then choose $M$ such that the average norm difference per point between subsequent steps is $\leq$ 0.05%. A Lorentzian kernel was chosen with a width of $\sigma = 0.025 E_{\mathrm{max}}$, where $E_{\mathrm{max}}$ is the largest energy calculated (compared with $\Delta E/E_i \approx$ 1-5% for modern time-of-flight spectrometers). The low energy regularization cutoff was chosen to be equal to the Lorentzian width $\sigma' = \sigma$. Neutron form factors and the dipole factor have been applied while the Bose correction factor has been omitted.

## 4.1 Quenched disorder

Disorder is ubiquitous in magnetic systems and takes many forms, including the presence of non-magnetic vacancies [52–55] and site-mixing [56, 57]. The presence of this disorder can play a role in the low energy dynamics even if it is confined to the non-magnetic sites [9, 58] due to the presence of spin-orbit or spin-lattice coupling. LSWT calculations of disordered systems are rare in the literature owing to the need to create a sufficiently large unit cell to capture the aperiodic nature of disorder. It should be noted that analytical approaches based on Green's functions exist but are complicated and restricted to the simplest systems with a large degree of approximation [59], with, in principle, an infinite number of self-consistency equations needing to be solved. Other recent approaches to treating disorder have involved real-space perturbation theory and Monte Carlo simulations [60]. Within standard LSWT, disorder can be captured by building a large magnetic unit cell. The momentum-space resolution is ultimately limited to the inverse linear size of the supercell, with artifacts arising due to the assumption of periodic repetition.

Motivated by recent work on rare-earth triangular lattice quantum spin liquid candidates, we now consider a $S = 1/2$ triangular lattice antiferromagnet model. The presence of chemical disorder in the non-magnetic environment manifests itself in a variation of the crystalline electric field, resulting in a varied $g$-tensor when projected onto the ground state doublet and anisotropic pseudo-dipolar exchange interactions [61]. Here, we only consider a variation of the $g$-factor and assume a Heisenberg exchange interaction for simplicity. We take a supercell of 30×30 spins and reduce the resolution to 30 $\mathbf{q}$-points per inverse lattice constant to eliminate artifacts from the periodic repetition of the unit cell. We apply a magnetic field along $z$,

$$\hat{\mathcal{H}} = J \sum_{\langle i,j \rangle} \hat{\mathbf{S}}_i \cdot \hat{\mathbf{S}}_j - h \sum_i g_\parallel(i) \hat{S}_i^z, \tag{42}$$

assuming the $g$-factors are normally distributed, $g_\parallel \sim N(\mu, \sigma^2)$. If the $g$-factor were not disordered ($\sigma = 0$), a single coherent gapped spin wave mode would be observed above the saturation field, $h_{sat} = 9JS/g_\parallel$, associated with the coherent precession of the spins about the magnetic field direction along which they are polarized. However, taking $\mu = 1.0$ and

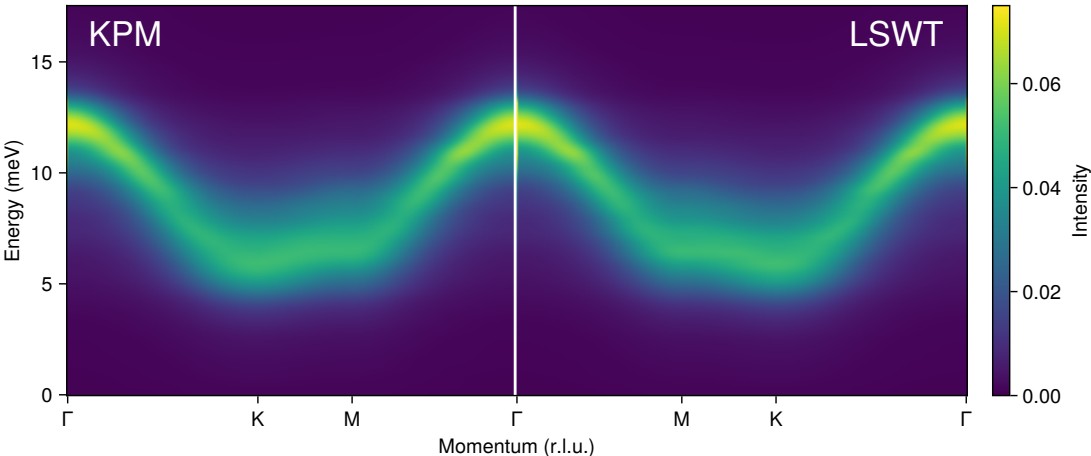

Figure 1: Comparison of the perpendicular dynamical spin structure factor calculated using KPM (left) and LSWT (right) for the $g$-factor disordered triangular lattice antiferromagnet.

$\sigma = \frac{1}{6}$, and setting $h = 2.5h_{sat}$, one observes considerable broadening of the field-saturated excitations (Fig. 1), as previously calculated in Ref. [62]. The truncation order to converge to within tolerance was $M = 75$. The distance between sampled **q**-points is 0.25 Å$^{-1}$ in the global frame, giving 249 **q**-points each for KPM and LSWT. For the results of Fig.1, the KPM calculation completed approximately 50 times faster than the traditional LSWT calculation (all benchmarks were taken using our preliminary implementation of KPM within the Sunny software package [63]). An alternatively linear scaling approach is to measure dynamical correlations within the classical Landau-Lifshitz dynamics. It is difficult to directly compare performance with KPM, because LLD requires additional averaging over thermal fluctuations, which are absent from the KPM calculation.

## 4.2   Incommensurability

The presence of frustrated interactions can lead to noncollinear magnetic structures where spins smoothly rotate with an associated magnetic ordering wavevector $|\boldsymbol{k}| = 2\pi/\lambda$ with a period of rotation, $\lambda$. Perhaps the simplest example of such a system is a one-dimensional chain with nearest-neighbor Heisenberg exchange, $J$, along with antisymmetric (or Dzyaloshinskii-Moriya) coupling [64,65] along the nearest-neighbor coupling direction, $\vec{D} = (D, 0, 0)$. Here frustration originates from the competition between the Heisenberg exchange which favors collinear alignment and antisymmetric exchange which cants neighboring spins. The classical energy of such a system is minimized when $\boldsymbol{k} = \arctan(D/J)$, with spins rotating uniformly in the $x - y$ plane. For $\boldsymbol{k} \neq p/q$, where $p$ and $q$ are coprime integers, the ordering wavevector is incommensurate with the lattice. That it to say, translational symmetry, which is broken by the magnetic order, cannot be restored by a redefinition of the unit cell. In the presence of continuous $SO(2)$ rotation symmetry, a transformation to a rotating frame can restore translational symmetry [23] allowing for a LSWT treatment. In the absence of spin rotational symmetry, umklapp terms destabilize the magnetic order making a rotating frame LSWT description impossible. At special points, a hidden $SO(2)$ symmetry may exist [66], but in general a large supercell must be created and the treatment is approximate. One example that evades a rotating frame treatment is the case of coupled zig-zag chains with opposite $\vec{D}$ (DMI) vectors. Such a situation is present in $\beta$-CaCr$_2$O$_4$, where the two-fold screw rotation symmetry between neighboring zig-zag chains stabilizes a ground state comprising an incommensurate structure

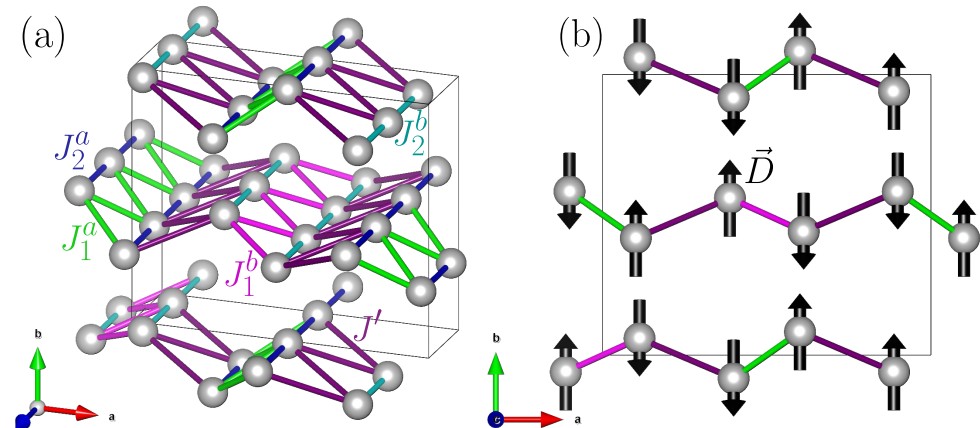

Figure 2: Incommensurate counterrotating zig-zag spiral model inspired by $CaCr_2O_4$. Plotted are the (a) Heisenberg exchanges and (b) DMI vector directions. Figure created using VESTA [67]. The .cif file was taken from Ref. [68].

of cycloids of opposite chirality [69–73]. Inspired by this compound we consider coupled zig-zags with opposite DMI vectors

$$\hat{\mathcal{H}} = \sum_{\langle i,j \rangle_\nu} J_\nu \hat{\mathbf{S}}_i \cdot \hat{\mathbf{S}}_j + \vec{D}_{ij} \hat{\mathbf{S}}_i \cdot \hat{\mathbf{S}}_j + \mu (\hat{S}_i^y)^2 . \tag{43}$$

The exchange parameters $J_\nu$ are labeled in Fig. 2a. With easy-plane anisotropy, the moments are confined to the $a-c$ plane. The staggering of the DM vector (Fig. 2b) on the legs of the zig-zag chains promotes a cycloidal magnetic structure with alternating chirality.

For simplicity, we assume that $J_{1a} = J_{1b} = J_1$, $J_{2a} = J_{2b} = J_2$, that $J_2, J' \gg J_1, D$ and the magnetic structure has a cylindrical envelope. The classical energy is then minimized for

$$\cos \frac{k_z}{2} = -\frac{J'}{4 J_2} . \tag{44}$$

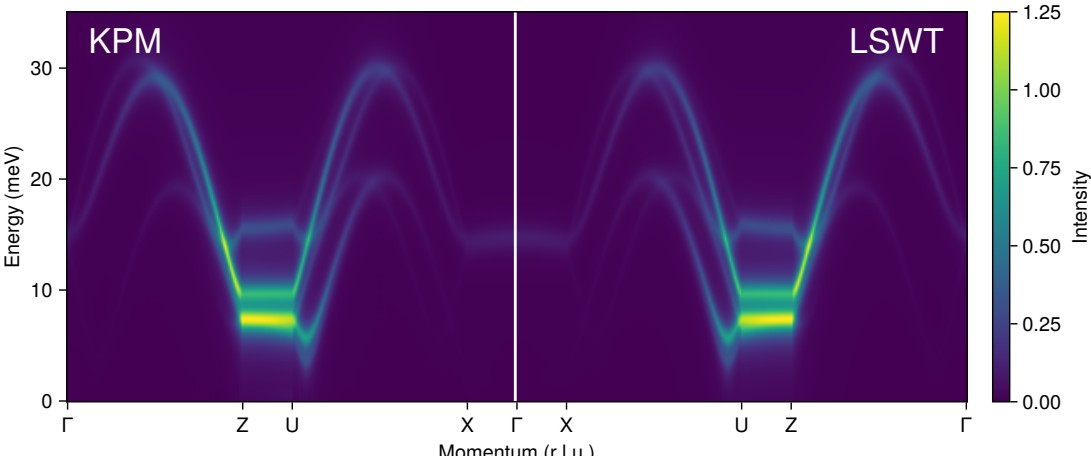

Figure 3: Comparison of the perpendicular dynamical spin structure factor calculated using KPM (left) and LSWT (right) for the counterrotating zig-zag model described in the main text.

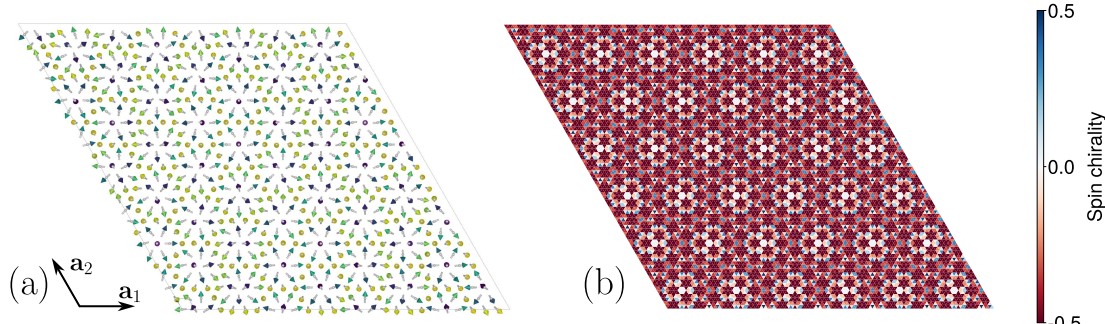

Figure 4: (a) Real space spin structure for the skyrmion lattice model described in the main text. Shown is a single unit cell, with dipoles colored based on $\langle \hat{S}_i^z \rangle$. (b) Scalar spin chirality $\mathbf{S}_1 \cdot (\mathbf{S}_2 \times \mathbf{S}_3)$ for each plaquette on the lattice. Nine magnetic unit cells are plotted. The color scale has been saturated to aid visibility.

With an antiferromagnetic $J_1$, we take $J_2 = 8J_1$ and $J' = -4J_1$, leading to an incommensurate propagation vector $\boldsymbol{k} = (0, 0, 0.0.53989)$. A small DMI, $\vec{D} = 0.5J_1$ sets the chirality. We take the nearby commensurate point $\boldsymbol{k} = (0, 0, 20/37)$ and approximate the cell as commensurate. The presence of finite $J_1$ gives rise to umklapp terms which, if large enough, tend to destabilize a single-$\boldsymbol{k}$ magnetic structure giving rise to multi-$\boldsymbol{k}$ order.

Figure 3 shows a comparison between the KPM calculation (left) and LSWT (right). The truncation order taken was $M = 175$. The distance between sampled $\mathbf{q}$-points is 0.03 Å$^{-1}$ in the global frame, giving 273 $\mathbf{q}$-points for each of the KPM and LSWT calculations. Despite the significantly smaller supercell size than the calculation in section 4.1, KPM still provides a speed up of approximately a factor of four, as benchmarked in our preliminary implementation [63]. Again, we remark that the unsuitability of the rotating frame LSWT formalism is common to many systems with $SO(2)$ symmetry breaking, for example Li$_2$IrO$_3$ [66, 74–77].

## 4.3 Multi-k

The existence of topologically non-trivial spin-textures in noncentrosymmetric systems with non-zero antisymmetric exchange has long been discussed [78–80]. Even in the absence of antisymmetric exchange, competition between symmetric exchange interactions can give rise to skyrmion lattice order [81–83]. These states are of great interest for their potential device applications [84]. Here we take the model of Ref. [83] and examine the dynamics in the triple-$\boldsymbol{k}$ phase using KPM. The Hamiltonian under consideration is

$$\hat{\mathcal{H}} = J_1 \sum_{\langle i, j \rangle}' \hat{\mathbf{S}}_i \cdot \hat{\mathbf{S}}_j + J_3 \sum_{\langle\langle\langle i, j \rangle\rangle\rangle} \hat{\mathbf{S}}_i \cdot \hat{\mathbf{S}}_j - h \sum_i \hat{S}_i^z,$$

with ferromagnetic nearest neighbor, $J_1$ and antiferromagnetic third-nearest-neighbor, $J_3$. The classical energy is minimized simultaneously for three wavevectors [83], reflecting the three-fold rotational symmetry on the triangular lattice. The wavevectors which minimize the classical energy satisfy [83]

$$|\boldsymbol{k}| = \frac{2}{a} \cos^{-1}\left[ \frac{1}{4}\left(1 + \sqrt{1 - \frac{2J_1}{J_3}}\right) \right]. \tag{45}$$

We select a point in the triple-$\boldsymbol{k}$ part of the phase diagram, stabilized by a finite field, $J_3 = -3J_1$, $h = 2.5J_3$ and $k_B T = 0.3J_3$. The wavevector that minimizes the classical energy is incommensurate with the lattice, leading to a magnetic unit cell lattice constant of $a_m = 3.26929a$. We

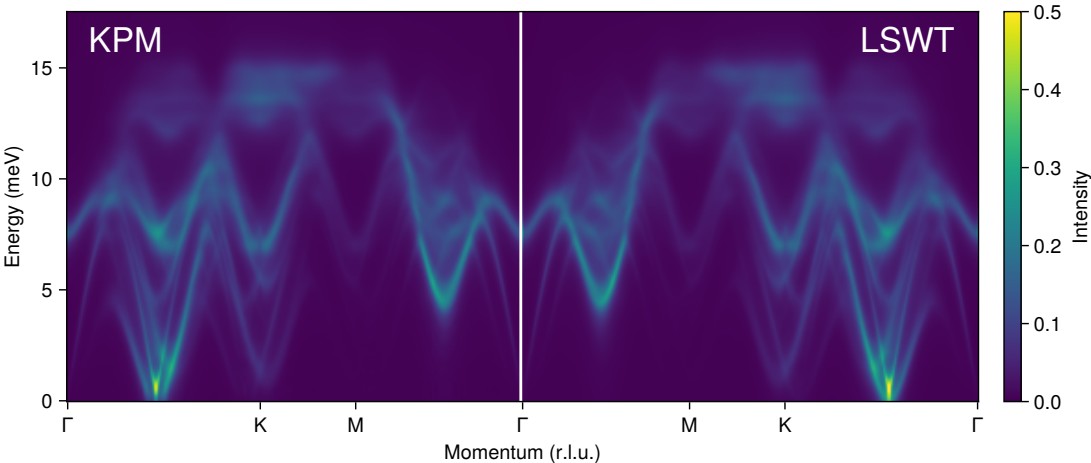

Figure 5: Comparison of the perpendicular dynamical spin structure factor calculated using KPM (left) and LSWT (right) for the skyrmion lattice model described in the main text.

can find a nearby commensurate point by rationalizing the lattice parameter as $a_m/a = 85/26$ and taking a $26 \times 26 \times 1$ dimensional unit cell. We note that the approximate commensurate system is extremely close to the incommensurate point, with $\frac{85}{26}/3.26929 = 0.999983$. The ground state (Fig. 4) was determined by simulated annealing, followed by gradient descent.

Figure 5 shows the spectrum calculated using KPM (left) and LSWT (right). The truncation order to converge to within tolerance was $M = 125$. The distance between sampled **q**-points is $0.25\,\text{Å}^{-1}$ in the global frame, giving 249 **q**-points for each of the KPM and LSWT calculations. A small diagonal positive shift of $\epsilon = 10^{-2}$ was added to the spectrum to remove a discontinuity at the Goldstone mode in both LSWT and KPM. For this model, the KPM calculation offers a speed up of approximately a factor of 15 compared with the conventional LSWT method. The KPM calculation faithfully captures the complicated overlapping dispersive modes present due to the large unit cell size, showing strong agreement with LSWT.

## 5 Computational complexity

The key algorithmic result of this paper is the favorable scaling of computational cost as a function of system size. A traditional LSWT calculation involves direct matrix diagonalization, at a cost that scales cubically with the volume of the magnetic unit cell. With KPM acceleration, the cost is reduced to linear scaling in system size. In this section, we quantify the speedup, as a function of system size, for a simple benchmark problem.

Consider, for simplicity, the model of an anisotropic antiferromagnetic chain of length $n$. Since each site carries a single boson, the size of the dynamical matrix $D$ appearing in Eq. (1) is then $2n \times 2n$. Traditionally, one would diagonalize this matrix directly, as in Eq. (6), to obtain the dynamical correlations of Eq. (15). The cost of direct diagonalization scales like $\mathcal{O}(n^3)$. By design, KPM avoids this expensive diagonalization operation; instead, KPM performs a sequence of $M$ sparse matrix-vector products via the recursion of Eqs. (35)–(39). The total cost of this recursion scales like $\mathcal{O}(nM)$, with a prefactor that depends sensitively on the sparsity of the matrix $D$. The polynomial order $M$ should be selected according to the required resolution in frequency $\omega$, relative to the total spectral bandwidth. Typically, $M$ will be order $10^2$.

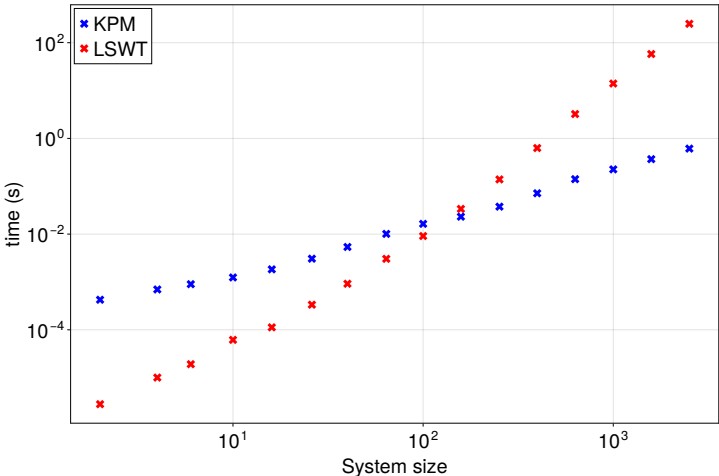

Figure 6: Scaling of numerical complexity of LSWT (red) compared with KPM (blue) for a linear chain of size $n$. In our benchmarking exercise KPM offers a clear advantage for unit cell sizes $\gtrsim 200$.

Figure 6 shows timings for the core operations described above, as a function of chain length $n$. All benchmarks were taken using a single thread on an Intel Core™ i7-1185G7 CPU. For the traditional LSWT timings, we benchmarked only the cost of the solving the generalized eigenvalue problem, Eq. (6), using the OpenBLAS implementation of the LAPACK subroutine ZHEGV. For KPM, we benchmarked only the costs of the Cheyshev recursion, Eqs. (35)–(39), multiplied by a factor of three. Therefore the KPM timings are reflective of the cost to evaluate the structure factor $\mathcal{S}^{\alpha,\beta}(\mathbf{q}, \omega)$ for all nine components $\alpha, \beta \in \{x, y, z\}$ at a single $\mathbf{q}$. We also selected $M = 350$, which yields a relatively high energy resolution. With these choices, KPM becomes clearly favorable over a traditional LSWT calculation for chain lengths $n \gtrsim 200$.

Some further technical remarks: (1) In this simple chain model, each site interacts only with its two nearest neighbors. If the number of exchange interactions were larger, then the cost of a matrix-vector multiply operation would proportionally increase. This, in turn, would slow KPM but not direct diagonalization. (2) Our benchmarks of KPM do not include the $m$ summation of Eq. (40) because this cost is independent of system size, and becomes relatively small beyond the crossover point, $n \gtrsim 200$. (3) With planned optimizations to the KPM implementation within Sunny [63], the KPM result may become about three times faster.

## 6 Conclusion

In this paper, we presented a method for calculating dynamical correlations using linear spin wave theory based on the kernel polynomial method. This method replaces the matrix diagonalization, inherent to conventional LSWT implementations, with a matrix approximation in terms of a Chebyshev expansion. This reduces computational complexity from $\mathcal{O}(N^3)$ to $\mathcal{O}(N)$ in the size $N$ of the magnetic unit cell. The utility of this method has been demonstrated by calculating the dynamical structure factor for three examples of systems with large system sizes. In all cases, the KPM shows high fidelity to the exact LSWT result, while significantly reducing computational time. It is suggested that this method will prove valuable in the inverse modeling of spectroscopic data, including inelastic neutron scattering, for systems which are incommensurate, multi-$\mathbf{k}$ or possess quenched disorder.

## Acknowledgments

The authors acknowledge useful conversations with D. Pajerowski and J. Rau.

**Funding information**    H. L. was supported by a Research Fellowship from the Royal Commission for the Exhibition of 1851. K. B. acknowledges support from the Department of Energy under Grant No. DE-SC-0022311, with a supplement provided by the Neutron Scattering program. The work at Georgia Tech (S. Quinn, C. D. Batista, M. Mourigal) was supported by U.S. Department of Energy, Office of Science, Basic Energy Sciences, Materials Sciences and Engineering Division under award DE-SC-0018660.

## A    Conditions for a canonical transformation

Assume an $n$-component vector of bosonic creation operators $\hat{\boldsymbol{\alpha}}^{\dagger}$. Following the notation of Sec. 2, it is convenient to collect these operators with their Hermitian conjugates into a single $2n$-component vector,

$$\hat{\mathbf{a}}^{\dagger} = (\hat{\boldsymbol{\alpha}}^{\dagger}, \hat{\boldsymbol{\alpha}}), \qquad \hat{\mathbf{a}} = \begin{pmatrix} \hat{\boldsymbol{\alpha}} \\ \hat{\boldsymbol{\alpha}}^{\dagger} \end{pmatrix}. \tag{A.1}$$

With this notation, the canonical commutation relations,

$$[\alpha_i, \alpha_j] = 0, \qquad [\alpha_i, \alpha_j^{\dagger}] = \delta_{ij}. \tag{A.2}$$

take the compact form,

$$[a_i, a_j^{\dagger}] = \tilde{I}_{ij}, \tag{A.3}$$

where the diagonal matrix $\tilde{I}$ has been defined in Eq. (5).

Now consider some linear transformation $\hat{\mathbf{b}} = S\hat{\mathbf{a}}$ where $S = T^{-1}$ as in Eq. (2). To allow the decomposition $\hat{\mathbf{b}}^{\dagger} = (\hat{\boldsymbol{\beta}}^{\dagger}, \hat{\boldsymbol{\beta}})$ for some $n$ operators $\hat{\boldsymbol{\beta}}^{\dagger}$, we require that $S$ has the structural form,

$$S = \begin{pmatrix} U & V \\ V^* & U^* \end{pmatrix}. \tag{A.4}$$

Of interest are the commutation relations following the linear transformation of Eq. (2),

$$[b_i, b_j^{\dagger}] = [(Sa)_i, (Sa)_j^{\dagger}]. \tag{A.5}$$

Using $(Sa)^{\dagger} = a^{\dagger}S^{\dagger}$, and bilinearity of the commutator, one finds

$$[b_i, b_j^{\dagger}] = S_{il}[a_l, a_m^{\dagger}]S_{mj}^{\dagger} \tag{A.6}$$

$$= (S\tilde{I}S^{\dagger})_{ij}. \tag{A.7}$$

That is, the new operators will satisfy bosonic commutation relations,

$$[b_i, b_j^{\dagger}] = \tilde{I}_{ij}, \tag{A.8}$$

if and only if $S$ obeys the para-unitary condition $S\tilde{I}S^{\dagger} = \tilde{I}$, or equivalently, $S^{-1} = \tilde{I}S^{\dagger}\tilde{I}$.

Canonical transformations are isomorphisms, so equivalent conditions can be imposed on the inverse transform. It follows that if $T$ has a structure analogous to Eq. (A.4) and is para-unitary, then the transformation $S = T^{-1}$ is canonical. Both conditions are satisfied by the eigen-decomposition of Eqs. (9) and (10) in the main text.

# B  Formulating dynamical correlations

When the many-body Hamiltonian $\hat{\mathcal{H}}$ reduces to a sum of *decoupled* harmonic oscillators, as in LSWT,

$$\hat{\mathcal{H}} = \sum_{j=1}^{n} E_j \left( \hat{\beta}_j^\dagger \hat{\beta}_j + \frac{1}{2} \right), \tag{B.1}$$

the expression of $C_\omega$ given in Eq. (13) can be simplified because it reduces to a sum of individual contributions from each harmonic oscillator. This is a direct consequence of the factorization of the partition function $\mathcal{Z} = \prod_j \mathcal{Z}_j$ into a product of partition functions $\mathcal{Z}_j = \sum_{n_j=0}^{\infty} e^{-\beta E_j(n_j+\frac{1}{2})}$ for each oscillator labelled by $j$. In other words, the eigenstates of $\hat{\mathcal{H}}$ can be expressed as a tensor product of eigenstates of each individual oscillator:

$$|\mathbf{n}\rangle = \otimes_j |n_j\rangle, \tag{B.2}$$

where

$$|n_j\rangle = \frac{[\hat{\beta}_j^\dagger]^{n_j}}{\sqrt{n_j!}} |0\rangle, \tag{B.3}$$

and $E_\mathbf{n} = \sum_j E_j(n_j + \frac{1}{2})$. We can then rewrite Eq. (13) as

$$C_\omega = \sum_{j=1}^{n} \sum_{n_j=0}^{\infty} \langle n_j | \hat{\mathcal{B}}^\dagger | n_j+1 \rangle \langle n_j+1 | \hat{\mathcal{A}} | n_j \rangle \frac{e^{-\beta E_j(n_j+1/2)}}{\mathcal{Z}_j} \delta(E_j - \omega), \tag{B.4}$$

where we have used the fact that $\hat{\mathcal{A}}$ and $\hat{\mathcal{B}}$ are linear in the operators $\hat{\beta}_j^\dagger$ and $\hat{\beta}_j$ and that we are only considering $\omega > 0$. Now we use

$$\langle n_j+1 | = \frac{\langle n_j | \hat{\beta}_j}{\sqrt{n_j+1}}, \tag{B.5}$$

to obtain

$$C_\omega = \sum_{j=1}^{n} \sum_{n_j=0}^{\infty} \frac{\langle n_j | \hat{\mathcal{B}}^\dagger \hat{\beta}_j^\dagger | n_j \rangle \langle n_j | \hat{\beta}_j \hat{\mathcal{A}} | n_j \rangle}{n_j+1} \frac{e^{-\beta E_j(n_j+1/2)}}{\mathcal{Z}_j} \delta(E_j - \omega). \tag{B.6}$$

Now we use

$$\langle n_j | \hat{\beta}_j \hat{\mathcal{A}} | n_j \rangle = \langle n_j | \hat{\beta}_j \mathbf{a}^\dagger | n_j \rangle \mathbf{u} = (1+n_j)(\mathbf{t}_j^\dagger \mathbf{u}) \delta_{ij},$$
$$\langle n_j | \hat{\mathcal{B}}^\dagger \hat{\beta}_j^\dagger | n_j \rangle = \langle n_j | \mathbf{a}\, \hat{\beta}_j^\dagger | n_j \rangle \mathbf{v}^\dagger = (1+n_j)(\mathbf{v}^\dagger \mathbf{t}_j) \delta_{ij}, \tag{B.7}$$

to obtain

$$C_\omega = \sum_{j=1}^{n} (\mathbf{v}^\dagger \mathbf{t}_j) \left[ \sum_{n_j=0}^{\infty} (n_j+1) \frac{e^{-\beta E_j(n_j+1/2)}}{\mathcal{Z}_j} \delta(E_j - \omega) \right] (\mathbf{t}_j^\dagger \mathbf{u}). \tag{B.8}$$

Finally, we use

$$\sum_{n_j=0}^{\infty} (n_j+1) \frac{e^{-\beta E_j(n_j+1/2)}}{\mathcal{Z}_j} = 1 + n(E_j), \tag{B.9}$$

with $n(E_j) = (e^{\beta E_j} - 1)^{-1}$ being the Bose function, to obtain the final $\omega > 0$ result,

$$C_\omega = [1 + n(\omega)] \sum_{j=1}^{n} (\mathbf{v}^\dagger \mathbf{t}_j) \delta(E_j - \omega)(\mathbf{t}_j^\dagger \mathbf{u}). \tag{B.10}$$

# C   Review of the kernel polynomial method

The kernel polynomial method (KPM) allows low-rank, stochastic approximation of matrix functions $f(A)$ provided that the eigenvalues of $A$ are real and bounded. Without loss of generality, we may assume that $A$ has been rescaled to have eigenvalues between $-1$ and $1$. KPM enables estimation of the matrix-vector products $f(A)\mathbf{u}$ without explicit construction of the matrix $f(A)$. This appendix reviews the procedure.

## C.1   Chebyshev expansion of a function

First, consider the function $f(\cdot)$ applied to a scalar $x$. (Think of $x$ as an arbitrary eigenvalue of $A$.) We will approximate $f(x)$ using a series expansion in Chebyshev polynomials. On the interval $-1 \le x \le 1$, the Chebyshev polynomials can be written

$$T_m(x) = \cos(m \arccos x). \tag{C.1}$$

Many results from Fourier analysis carry over to Chebyshev series. For example, the cosines are orthonormal,

$$\int_0^\pi \cos(m\phi)\cos(m'\phi)d\phi = q_m \delta_{m,m'}, \tag{C.2}$$

where

$$q_m = \begin{cases} \pi, & m = 0, \\ \pi/2, & m \ge 1. \end{cases} \tag{C.3}$$

Upon changing the integration variable, $x = \cos\phi$, we find orthonormality of Chebyshev polynomials,

$$\int_{-1}^{+1} T_m(x)T_{m'}(x)w(x)dx = q_m \delta_{m,m'}. \tag{C.4}$$

The appropriate weight function,

$$w(x) = (1 - x^2)^{-1/2}, \tag{C.5}$$

follows from $d\phi/dx = -w(x)^{-1}$.

Chebyshev polynomials are complete on the interval $-1 \le x \le 1$, allowing expansion of an arbitrary function,

$$f(x) = \sum_{m=0}^{\infty} c_m T_m(x). \tag{C.6}$$

By orthonormality, the expansion coefficients are

$$c_m = \frac{1}{q_m} \int_{-1}^{+1} w(x)T_m(x)f(x)dx. \tag{C.7}$$

When $f(x)$ is a smooth function, Chebyshev-Gauss quadrature is effective, as described in Appendix C.2.

An approximate polynomial expansion is obtained by restricting the series to some finite polynomial order $m < M$. If $f(x)$ includes singularities, then naïve truncation may lead to "ringing" artifacts (Gibbs oscillations). These artifacts can be eliminated in a general way by introducing damping coefficients $g_m^M$ that decrease with $m$,

$$f(x) \approx \sum_{m=0}^{M-1} g_m^M c_m T_m(x). \tag{C.8}$$

A common choice of coefficients,

$$g_m^M = \frac{(M - m + 1)\cos\frac{\pi m}{M+1} + \sin\frac{\pi m}{M+1}\cot\frac{\pi}{M+1}}{M + 1}, \tag{C.9}$$

is derived from the "Jackson kernel" [30], which originates from Fourier analysis [85, 86].

In our applications to LSWT, we will be working with a smoothed function $f(x)$. If $M$ is sufficiently large to resolve all the sharp features in $f(x)$, then Gibbs oscillations will be controlled, and it becomes preferable to select

$$g_m^M = 1, \tag{C.10}$$

which reduces the error of the polynomial approximation.

## C.2 Coefficients via the discrete cosine transform

To numerically estimate the coefficients $c_m$ of Eq. (C.7) for smooth $f(x)$, one may use Chebyshev-Gauss quadrature,

$$c_m \approx \frac{\pi}{q_m N} \sum_{n=0}^{N-1} T_m(x_n) f(x_n). \tag{C.11}$$

The $N$ abscissas of integration are,

$$x_n = \cos\left[\frac{\pi}{N}(n + 1/2)\right]. \tag{C.12}$$

For the KPM approximation of Eq. (C.8), one requires coefficients $c_0 \ldots c_{M-1}$, and a reasonable quadrature order is $N \gtrsim 2M$.

The Chebyshev polynomials are defined to satisfy $T_m(\cos\phi) = \cos(m\phi)$. It follows,

$$T_m(x_n) = \cos\left[\frac{\pi}{N}(n + 1/2)m\right]. \tag{C.13}$$

Then Eq. (C.11) becomes

$$c_m \approx \frac{\pi}{q_m N} \tilde{f}_m, \tag{C.14}$$

where

$$\tilde{f}_m = \sum_{n=0}^{N-1} f(x_n) \cos\left[\frac{\pi}{N}(n + 1/2)m\right], \tag{C.15}$$

is a discrete cosine transform of the second kind (DCT-II). Given data for $f(x_n)$, all coefficients $c_0 \ldots c_{M-1}$ can be evaluated in a single shot at cost $\mathcal{O}(N \ln N)$ using a package like FFTW [87].

## C.3 Fast matrix-vector products

The approximation of Eq. (C.8) also applies to functions of a diagonalizable matrix $A$,

$$f(A) \approx \sum_{m=0}^{M-1} g_m^M c_m T_m(A), \tag{C.16}$$

provided that the eigenvalues of $A$ are real and within the range $[-1, 1]$.

The Chebyshev matrix-polynomials can be calculated using a two-term recurrence,

$$T_{m+1}(A) = 2A T_m(A) - T_{m-1}(A), \tag{C.17}$$

starting from $T_0 = I$ and $T_1 = A$.

Frequently, only a matrix-vector product $f(A)\mathbf{u}$, for some vector $\mathbf{u}$, is required. In this context, it is advantageous to avoid explicit construction of the matrix $f(A)$. Instead, we may calculate

$$f(A)\mathbf{u} \approx \sum_{m=0}^{M-1} g_m^M c_m \boldsymbol{\varphi_m}\,, \tag{C.18}$$

where the vectors $\boldsymbol{\varphi}_m = T_m(A)\mathbf{u}$ are generated through the recurrence,

$$\boldsymbol{\varphi}_{m+1} = 2A\boldsymbol{\varphi}_m - \boldsymbol{\varphi}_{m-1}\,, \tag{C.19}$$

starting from $\boldsymbol{\varphi}_0 = \mathbf{u}$ and $\boldsymbol{\varphi}_1 = A\mathbf{u}$. Typically the matrix-vector product $A\boldsymbol{\varphi}_m$ will be very efficient to evaluate (e.g., $A$ will be sparse), yielding an overall linear scaling cost with system size.

## C.4 Stochastic approximation

KPM is frequently used in conjunction with stochastic approximation, e.g.,

$$f(A) \approx [f(A)\mathbf{r}]\mathbf{r}^\dagger\,, \tag{C.20}$$

where $\mathbf{r}$ is a random vector. The approximation is unbiased if $\langle \mathbf{r}\mathbf{r}^\dagger \rangle = I$. That is, the random components $r_i$ should be independent, with zero mean and unit variance.

Stochastic approximation enables efficient evaluation of matrix-vector products, $f(A)\mathbf{u} \approx [f(A)\mathbf{r}](\mathbf{r}^\dagger\mathbf{u})$. Crucially, one can precompute the matrix vector product $f(A)\mathbf{r}$ independently of $\mathbf{u}$ via the Chebyshev expansion of Eq. (C.18). Then, for each $\mathbf{u}$ of interest, the vector dot product $\mathbf{r}^\dagger\mathbf{u}$ is relatively fast to evaluate.

Matrix elements approximated as in Eq. (C.20) will tend to have large stochastic error. To reduce it, one can average over multiple random vectors. Stacking these vectors into the columns of a matrix $R$, the stochastic approximation may be written $f(A) \approx f(A)RR^\dagger$, and is unbiased if $\langle RR^\dagger \rangle = I$. This viewpoint suggests opportunities for reducing stochastic error [88]. Specifically, the $R$ matrix may be designed in a way that leverages of the decay of matrix elements $f(A)_{ij}$ as a function of distance between sites $i$ and $j$ [89]. Further reduction in stochastic error is possible by estimating $f(A) \approx (d/dA^T)\mathrm{tr}\, g(A)RR^\dagger$, where $g(\cdot)$ is an antiderivative of $f(\cdot)$ viewed as a scalar function [90]. The key observation is that $g(\cdot)$ will typically be smoother than $f(\cdot)$, and this in turn leads to more sparsity of the matrix $g(A)$.

One might envision future applications of stochastic approximation to LSWT. For example, the dynamical matrix $D$ might formulated in real-space, for a large magnetic unit cell. Then a single stochastic approximation to the matrix function $f_\omega(\tilde{I}D)$ could be used to efficiently estimate structure factor data $\mathcal{S}(\mathbf{q}, \omega)$ for a large number of commensurate wavevectors $\mathbf{q}$. A very different approach is to use KPM to estimate the real-time evolution of dynamical observables, and employ stochastic approximation to the matrix trace [91].

# D Direct expansion of the bare intensities

In the main text, KPM was used to approximate the smoothed intensities $\tilde{C}_\omega$. Here we explore an alternative approach: direct expansion of the bare intensity distribution $C_\omega$. A benefit of this approach is that the $m$-summation of Eq. (40) can be replaced with a faster convolution operation. A disadvantage, however, is that the resulting line-broadened approximation converges more slowly in the polynomial order $M$.

Recall from Eq. (17) that $C_\omega$ is expressible in terms of the distribution-valued matrix $\rho_+(\omega)$ of Eq. (25). Given our focus on positive frequencies $\omega > 0$, we have

$$\rho_+(\omega) = \delta(\omega - \tilde{I}D)\tilde{I}. \tag{D.1}$$

As before, let $\gamma$ be a scaling such that $A = \tilde{I}D/\gamma$ has eigenvalues within $-1$ and $1$. Then,

$$\rho_+(\omega) = f_\omega(A)\tilde{I}, \tag{D.2}$$

where $f_\omega(x) = (1/\gamma)\delta(\omega/\gamma - x)$ is suitable for Chebyshev expansion, Eq. (C.8). This yields

$$\rho_+(\omega) \approx \sum_{m=0}^{M-1} g_m^M c_{m,\omega} T_m(A)\tilde{I}. \tag{D.3}$$

The coefficients

$$c_{m,\omega} = \frac{1}{q_m \gamma} w(\omega/\gamma) T_m(\omega/\gamma), \tag{D.4}$$

are obtained by integrating Eq. (C.7) with $f(x) \to f_\omega(x)$ defined above. Definitions of $T_m(\cdot)$, $q_m$, and $w(\cdot)$ are provided in Eqs. (C.1)–(C.5). In the present context, the damping coefficients $g_m^M$ play a crucial role. Because the Dirac-$\delta$ is highly singular, naïvely setting $g_m^M = 1$ would lead to severe ringing at any truncation order $M$. Instead, one may select $g_m^M$ as in Eq. (C.9) to eliminate all ringing. Upon doing so, the polynomial approximation to $f_\omega(x)$ is non-oscillatory, with a width that decays like $1/M$.

Substitution of Eq. (D.3) into Eq. (17) yields a KPM approximation to the bare intensities,

$$C_\omega \approx [1 + n_{\mathrm{B}}(\omega)] \sum_{m=0}^{M-1} g_m^M c_{m,\omega} \mu_m, \tag{D.5}$$

where $\mu_m$ are the same Chebyshev moments as defined in the main text, Eq. (39). As before, these can be evaluated efficiently using sparse matrix-vector products, Eq. (38).

The remaining $m$-summations can be efficiently evaluated for a carefully selected set of frequencies,

$$\omega_n = \gamma x_n = \gamma \cos\left[\frac{\pi}{N}(n + 1/2)\right], \tag{D.6}$$

with $n = 0, 1, \ldots, N-1$. Note that $x_n$ previously appeared as the abscissas of Chebyshev-Gauss quadrature, Eq. (C.12). Substituting from Eq. (C.3) and Eq. (C.1), the bare intensities are,

$$C_{\omega_n} \approx [1 + n_{\mathrm{B}}(\omega_n)] \frac{w(x_n)}{\pi \gamma} \left\{ \mu_0 g_0^M + 2 \sum_{m=1}^{N-1} \mu_m g_m^M \cos\left[\frac{\pi}{N}(n + 1/2)m\right] \right\}. \tag{D.7}$$

To extend the $m$-summation, we have zero-padded the moment data by setting $\mu_m = 0$ for $m \geq M$.

The right term in brackets is a discrete cosine transform of the third kind (DCT-III) on the data $\{\mu_m g_m^M\}$. It can be efficiently evaluated, for all $n$ simultaneously, using a package such as FFTW. Note that the DCT-III is the inverse of the DCT-II operation, which appeared in Eq. (C.15).

Assuming all the moment data $\mu_m$ is available, the remaining computational cost to estimate $C_\omega$ at all frequencies $\omega \in \{\omega_0, \ldots, \omega_{N-1}\}$ scales like $\mathcal{O}(N \ln N)$. This makes it feasible select a large $N$ value (relative to $M$), evaluate the sums of Eq. (D.7), and then interpolate $C_\omega$ onto any desired set of frequencies.

Typically, the broadened intensities $\tilde{C}_\omega$ of Eq. (26) will be a convolution over $C_\omega$. In this case, one can estimate $C_\omega$ at regular intervals $\omega = \{0, \Delta\omega, 2\Delta\omega, \dots\}$ and use FFT acceleration to calculate $\tilde{C}_\omega$ via the convolution theorem, thereby avoiding the explicit $m$-summations of Eq. (40).

As mentioned in the beginning of this appendix, this scheme has a significant disadvantage: Direct KPM approximation to $C_\omega$ converges relatively slowly in the polynomial order $M$. Effectively, KPM must find a smoothed approximation to the Dirac-$\delta$, and its width will scale like $\gamma/M$. This relatively slow decay, of order $1/M$, then propagates to the errors in the estimates of the broadened intensities, $\tilde{C}_\omega$.

Conversely, in the main text, we applied KPM approximation to the already broadened intensities $\tilde{C}_\omega$ via Eq. (40). There, convergence is exponentially fast in $M$ once the KPM resolution $\gamma/M$ exceeds the characteristic scale of line broadening.

It is interesting to observe that the KPM approximation to either $C_\omega$ or $\tilde{C}_\omega$ is built upon identical Chebyshev matrix polynomials $T_m(A)$, and from these, identical Chebyshev moments $\mu_m$. The crucial difference, it seems, is that the direct approximation $C_\omega$ must introduce damping coefficients $g_m^M$ to avoid ringing artifacts in the approximate Dirac-$\delta$, prior to broadening.

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
