# Peer review of "Kernel Polynomial Method for Linear Spin Wave Theory"

_SciPost Physics, doi:SciPost Phys. 17, 145 (2024)_

## Round 2 · Referee Report · Anonymous (Referee 1) · 2024-3-20

# Report on "Kernel Polynomial Method for Linear Spin Wave Theory"

March 20, 2024

Dear Editor,

The authors describe a method of calculating spin wave spectra for Inelastic neutron scattering experiments, especially designed for large systems and unit cells and where explicit diagonalization is not needed. This paper is very well written, clear and well documented. I would definitely recommend its publication. Below are my comments:

- In addition to Ref [21-25], the authors may want to cite SpinWave, a software dedicated to spin wave simulations, S. Petit and F. Damay, Neutron News, Volume 27, Issue 4 (pp 27-28) (2016).

- Eq. 23 is quite complicated as arguments of Heaviside and Dirac functions are matrices (if I understand correctly). This could be pointed out in the text, for the sake of clarity.

- Eq. 25 - 27 are quite confusing, mixing properties of $C_{\omega'}$, with those imposed on $G(\omega, \omega')$. It also seems that the integration over $\omega'$ has vanished in obtaining Eq. 26. Why should $\theta(x)$ mask negative values of $x$ since $C(\omega)$ already does ? This part is definitely not very clear to me. Some additional explanations are needed welcome.

- Eq. 29 is also quite complicated, especially because the method needs a proper value of $\gamma$, which is, seemingly, obtained from much more complex methods. It is difficult to figure out wheter this part makes the method much more complex or not, for anybody who would like to implement it. Could the authors comment about this ?

- The authors explain that the KPM method becomes more efficient than conventional LSWT for system size above $\sim 200$ (Figure 6). It seems this corresponds to quite large systems, with very likely complicated magnetic structures. I thus have two questions : given these larges unit cells, does it remain possible to fit the exchanges couplings ? Furthermore, what about the mean field step inherent to SW based methods. Is it reasonable for such sizes ? Does it still converge properly ?

---

## Round 2 · Referee Report · Anonymous (Referee 2) · 2024-4-11

Strengths

The paper considers application of kernel polynomial method for calculation of linear response functions (in particular, dynamic susceptibilities) within the linear spin-wave theory.

Weaknesses

The paper is a bit technical, but at the same time provides an important contribution for practical implementation of spin-wave theory for complex systems.

Report

The authors suggest using kernel polynomial method for calculation of the dynamical spin correlation functions within the linear spin-wave theory. For systems with large unit cells straightforward application of the linear spin wave theory (LSWT) for calculation of dynamic susceptibilities may meet computational difficulties. The authors argue that their method is superior in comparison to LSWT with respect to the computational time. They demonstrate applicability of the method on 3 examples: quenched disorder, incommensurate magnetic order and the so called 'triple k' phase of skyrmion lattice order. The examples they analyse are rather impressive and provide demonstration of the power of their method. Therefore, in my opinion, the paper provides substantial contribution in the field of magnetism of insulating systems.

Requested changes

-

Recommendation

Publish (surpasses expectations and criteria for this Journal; among top 10%)

---

## Round 3 · Referee Report · Anonymous (Referee 2) · 2024-8-29

Strengths

The paper considers application of kernel polynomial method for calculation of linear response functions (in particular, dynamic susceptibilities) within the linear spin-wave theory. It shows that large unit cells can be treated within the proposed method and shows its applicability on concrete examples.

Weaknesses

-

Report

The authors have modified the paper according to the suggestions of one of the Referees. In my opinion the paper fulfills the criteria of SciPost Physics and can be published.

Requested changes

-

Recommendation

Publish (surpasses expectations and criteria for this Journal; among top 10%)

---

## Round 3 · Author Response

We thank the referee for their careful review of our manuscript and their thoughtful and insightful comments. Please find below our responses to each comment in turn.

  1. In addition to Ref [21-25], the authors may want to cite SpinWave, a software dedicated to spin wave simulations, S. Petit and F. Damay, Neutron News, Volume 27, Issue 4 (pp 27-28) (2016). A: We thank the referee for pointing out our oversight in referencing this software package. We have now added this reference to the manuscript.

  2. Eq. 23 is quite complicated as arguments of Heaviside and Dirac functions are matrices (if I understand correctly). This could be pointed out in the text, for the sake of clarity. A: We have updated the manuscript to include additional equations to better explain the steps from (21)–(25), yielding the compact functional representation of ρ+. We have also added a footnote to indicate that gω(·) is in reality a matrix-valued distribution, but for purposes of the formal manipulations, it is valid to assume some vanishingly small regularization which makes it an ordinary matrix function.

  3. Eq. 25 - 27 are quite confusing, mixing properties of C(ω′) , with those imposed on G(ω, ω′). It also seems that the integration over ω′ has vanished in obtaining Eq. 26. Why should θ(x) mask negative values of x since C(ω) already does? This part is definitely not very clear to me. Some additional explanations are needed welcome. A: We have replaced the text ”after a short calculation” with explicit steps leading to the final result [now labeled Eq. (29)]. We have also added a footnote to outline an alternate derivation which would avoid subtleties with the matrix-valued Dirac-δ.

  4. Eq. 29 is also quite complicated, especially because the method needs a proper value of γ, which is, seemingly, obtained from much more complex methods. It is difficult to figure out whether this part makes the method much more complex or not, for anybody who would like to implement it. Could the authors comment about this ? A: The scaling of the matrix was performed by the Lanczos algorithm. Simple implementations of this are available in a number of publicly available packages. The added complexity here comes from the non-hermiticity of the matrix, A. The extension to cover non-hemitian matrices is outlined in Refs. 46 and 47 and amounts to a few extra steps per iteration. In principle, the Lanczos step can be avoided if the approximate energy scales of the problem are known and hence a “safe” value of γ is known a priori. However, for implementation in Sunny.jl we wanted a robust way of finding this for any problem.

  5. The authors explain that the KPM method becomes more efficient than conventional LSWT for system size above ∼ 200 (Figure 6). It seems this corresponds to quite large systems, with very likely complicated magnetic structures. I thus have two questions: given these larges unit cells, does it remain possible to fit the exchanges couplings ? Furthermore, what about the mean field step inherent to SW based methods. Is it reasonable for such sizes ? Does it still converge properly. A: This is a very important question. As the referee points out, the challenge of forward calculation comprises two parts: the statistical mechanical question of finding the mean field ground state around which to expand, and then the subsequent challenge of describing the dynamics. Both of these steps depend on the exchange parameters and so any fitting process should perform both aspects of this. For a given set of parameters, the first step of finding a mean field ground state can be achieved using gradient descent, or Monte Carlo/Langevin dynamics methods, as performed in the paper. Sunny.jl includes a number of powerful tools to do this. In the models presented in this paper, this could be achieved without too much issue, so long as an appropriate supercell size was chosen. For a subset of models, approaches such as Luttinger-Tisza may be successful, and in the case of both the skyrmion lattice example and the counterrotating spiral, there are analytical expressions for the classical ground state wavevector, that agree with our numerical results. With respect to the question of fitting the exchanges for a large system size, we expect similar challenges to those found when fitting smaller system sizes. For both large and small systems, there is always the problem that, by changing the exchanges, one can destabilize the ground state meaning the starting state is no longer an energy minimum and the chosen supercell can no longer accommodate the mean field magnetic structure. It is therefore necessary to restrict oneself to the parameter space for which your unit cell permits the correct magnetic structure, or else choose a sufficiently large unit cell that can accommodate structures with a large number of periodicities. In this sense, fitting large unit cell sizes may provide an advantage, in the sense that we can allow the fitting procedure to pass through regions of parameter space where the wavelength becomes long. We also note that the spectra presented in this manuscript show a number of features which depend heavily on the parameters of the model which would aid in the determination of exchange constants.

In addition to the comments of the referees, we have made the following minor changes. 1. In Sect. 4.2 we quoted the ordering wavevector k = (0, 0, 460107). We have changed this to the wavevector of k′ = (0, 0, 0.53989) which is equivalent up to a shift of a reciprocal lattice vector k′ = (0, 0, 1) − k. Ultimately, the appropriate wavevector depends on the choice of the crystal unit cell, which we have not presented in the paper as this extra detail is extraneous. However, the choice of k′ is more appropriate and agrees with the convention of Eqn. 44. 2. We spotted a typographic error in the quoting of truncation order. For the skyrmion lattice example it should be M = 125 not M = 150, and for the incommensurate example it should be, M = 175 not M = 275. 3. We recomputed the approximated speed up compared with conventional LSWT quoted for the three models that we present in the manuscript. Due to some performance improvements in the LSWT code, these values have changed slightly compared with our initial tests. We have updated the values in the text to reflect the optimized LSWT code that is currently available in Sunny.jl. 4. The approximate speed up compared with LSWT was not quoted for the Skyrmion lattice example. We have therefore added the following sentence: “For this model, the KPM calculation offers a speed up of approximately a factor of 15.” 5. Finally, we have regenerated the figures to be sure that the output is representative of the values quoted in the text. The figures are visibly equivalent to the originals.

---

## Round 3 · List of Changes

• We have added the reference: SpinWave, a software dedicated to spin wave simulations, S. Petit and F. Damay, Neutron News, Volume 27, Issue 4 (pp 27-28) (2016)
• We have updated the manuscript to include additional equations to better explain the steps from (21)–(25), yielding the compact functional representation of ρ+. We have also added a footnote to indicate that gω(·) is in reality a matrix-valued distribution, but for purposes of the formal manipulations, it is valid to assume some vanishingly small regularization which makes it an ordinary matrix function.
• We have replaced the text “after a short calculation” with explicit steps leading to the final result [now labeled Eq. (29)]. We have also added a footnote to outline an alternate derivation which would avoid subtleties with the matrix-valued Dirac-δ.
• In Sect. 4.2 we quoted the ordering wavevector k = (0, 0, 460107). We have changed this to the wavevector of k′ = (0, 0, 0.53989) which is equivalent up to a shift of a reciprocal lattice vector k′ = (0, 0, 1) − k. Ultimately, the appropriate wavevector depends on the choice of the crystal unit cell, which we have not presented in the paper as this extra detail is extraneous. However, the choice of k′ is more appropriate and agrees with the convention of Eqn. 44.
• We corrected a typographic error in the quoting of truncation order. For the skyrmion lattice example it should be M = 125 not M = 150, and for the incommensurate example it should be, M = 175 not M = 275.
• We recomputed the approximated speed up compared with conventional LSWT quoted for the three models that we present in the manuscript. Due to some performance improvements in the LSWT code, these values have changed slightly compared with our initial tests. We have updated the values in the text to reflect the optimized LSWT code that is currently available in Sunny.jl.
• The approximate speed up compared with LSWT was not quoted for the Skyrmion lattice example. We have therefore added the following sentence: “For this model, the KPM calculation offers a speed up of approximately a factor of 15.”
• We have regenerated the figures to be sure that the output is representative of the values quoted in the text. The figures are visibly equivalent to the originals.

---

## Editorial Decision

published